# Potential of Rejected Sago Starch as a Coating Material for Urea Encapsulation

**DOI:** 10.3390/polym15081863

**Published:** 2023-04-13

**Authors:** Rajan Kavitha, Omar Latifah, Osumanu Haruna Ahmed, Primus Walter Charles, Kasim Susilawati

**Affiliations:** 1Department of Crop Science, Faculty of Agricultural Science and Forestry, Universiti Putra Malaysia, Bintulu Sarawak Campus, Bintulu 97008, Malaysia; 2Institute of Ecosystem Science Borneo, Universiti Putra Malaysia, Bintulu Sarawak Campus, Bintulu 97008, Malaysia; 3Faculty of Agriculture, University Sultan Sharif Ali Brunei, Kampus Sinaut, Km 33, Jalan Tutong, Kampung Sinaut, Tutong TB1741, Brunei; 4Department of Science and Technology, Faculty of Humanities, Management and Science, Universiti Putra Malaysia, Bintulu Sarawak Campus, Bintulu 97008, Malaysia; 5Department of Land Management, Faculty of Agriculture, Serdang 43400, Malaysia

**Keywords:** coating materials, efficient uptake, nutrient retention, slow-release mechanism

## Abstract

Increases in food production to meet global food requirements lead to an increase in the demand for nitrogen (N) fertilizers, especially urea, for soil productivity, crop yield, and food security improvement. To achieve a high yield of food crops, the excessive use of urea has resulted in low urea-N use efficiency and environmental pollution. One promising alternative to increase urea-N use efficiency, improve soil N availability, and lessen the potential environmental effects of the excessive use of urea is to encapsulate urea granules with appropriate coating materials to synchronize the N release with crop assimilation. Chemical additives, such as sulfur-based coatings, mineral-based coatings, and several polymers with different action principles, have been explored and used for coating the urea granule. However, their high material cost, limited resources, and adverse effects on the soil ecosystem limit the widespread application of urea coated with these materials. This paper documents a review of issues related to the materials used for urea coating and the potential of natural polymers, such as rejected sago starch, as a coating material for urea encapsulation. The aim of the review is to unravel an understanding of the potential of rejected sago starch as a coating material for the slow release of N from urea. Rejected sago starch from sago flour processing is a natural polymer that could be used to coat urea because the starch enables a gradual, water-driven mechanism of N release from the urea–polymer interface to the polymer–soil interface. The advantages of rejected sago starch for urea encapsulation over other polymers are that rejected sago starch is one of the most abundant polysaccharide polymers, the cheapest biopolymer, and is fully biodegradable, renewable, and environmentally friendly. This review provides information on the potential of rejected sago starch as a coating material, the advantages of using rejected sago starch as coating material over other polymer materials, a simple coating method, and the mechanisms of N release from urea coated with rejected sago starch.

## 1. Introduction

In plants, nitrogen (N) deficiency leads to symptoms such as stunted growth, yellowish leaves (chlorosis), and a thin and spindly stem [1]. Nitrogen is mobile within the crops; its deficiency can be found at tip of the leaf and moves along the middle towards the base [1]. In N-deficient crops, the protein content is low and the sugar content is high because there is insufficient N to combine with all the C chains stemming from the sugars that are then used to make proteins. It is prudential to ensure high yields and a high nitrogen use efficiency (NUE) in crop production systems. Nitrogen use efficiency is reflected by the ratio of the N assimilated in plant tissues to the total input of N fertilizer applied in the soil. In agronomic research, various indices are used to assess the efficiency of applied N [2]. In field studies, the indices are calculated based on differences in the crop yield and total N uptake with the aboveground biomass between fertilized and unfertilized plots. According to Dobermann [3], the agronomic framework is useful for understanding the factors governing N uptake and the NUE, to compare the short-term NUE in different environments, and to evaluate different N management strategies or technologies. The different method is simple, cost efficient, and suitable for NUE determination in field research [3]. When the values of NUE in crop yields were decreased at higher N rates, the findings indicated that the plants were unable to absorb N; this could be due to the saturation of plant absorption mechanisms. The determination of the NUE in crop plants is an important approach to evaluating the fate of the applied N fertilizers and their role in improving crop yields. 

Changing the rate of urea dissolution in the soil to synchronize the availability of inorganic N with crop requirements could be achieved by coating the urea to ascertain an improvement in the NUE. Moreover, owing to the fast release of N from the urea and the N becoming readily available in the soil, not all the N is used by plants due to factors such as the plants not efficiently absorbing 40 to 70% of the N applied from urea, which results in a low N uptake and NUE [4]. Consequently, the repeated or frequent application of urea to meet crop requirements not only causes an increased cost of crop production but also pollutes the environment and is associated with an imbalance in the amount of N that is then subjected to loss through nitrate (NO_3_^−^) leaching, ammonia (NH_3_) volatilization, and denitrification [5,6]. The slow-release technique from encapsulated urea is one of the appropriate methods for overcoming the rapid loss of N from urea and to provide available N for plant uptake and soil N nourishment. Bortoletto-Santos [7] suggested that the solubility of urea in water is reduced by coating it with suitable materials that slow the release of N in the soil. The mechanism involves a coated film physically creating a barrier during dissolution, which determines the dissolution rate in water when the coated urea is applied to the soil [8]. According to Thind et al. [9], the encapsulated urea prills lower the discharge rate of N in the field, which can result in lessening the leaching of harmful NO_3_^−^ into the groundwater and hindering the volatilization of NH_3_ into the atmosphere. In another related study on encapsulated urea, Mulder et al. [10] revealed that the physical characteristics of the material used to coat the urea governed the N release process in the soil. The mechanisms of N release from urea are very important, and the way that the release of N is regulated and how it acts with a coating when it is applied in the field should be well-understood [10].

The advantages of coated urea over conventional urea are cost savings and a reduced application frequency in the case of short-term crops, such as leafy vegetables, because less urea is applied to sufficiently provide the N required for the growth season when N is available in the soil for crop uptake [11]. Coated urea could also inhibit N loss, seed toxicity, hazardous emissions, and leaf burning [12]. In addition, coated urea improves the soil quality [13] and seed germination rates [12]. However, the materials used for urea coating must be considered, particularly with respect to their degradability rate. Despite the fact that sulfur-coated urea reduces the dissolution rate of the urea particles and imparts slow-release characteristics, Saleh and Hemati [14] stated that most of the well-established urea coating materials such as sulfur, phosphoric acids, and synthetic polymers are non-degradable, costly, and unfriendly to the environment. In addition, the problems associated with sulfur-coated urea are a high production cost, inconsistent release patterns, non-uniform coating thickness, unwanted surface cracks, essential abrasion, poor impact resistance, and processing complexity [15,16]. Owing to these problems, studies exploring the use of natural polymers for urea coatings have been widely conducted.

## 2. Natural Polymers for Encapsulating Urea

The materials used for coating urea can be divided into organic and inorganic coatings, as summarized in Figure 1, with the aim of controlling the fast release of ordinary or uncoated urea. The rapid release of N from urea is not necessarily good because a too-high concentration of N released into the soil might cause fertilizer burn to the crop, the contamination of ground water due to NO_3_^−^ leaching, and N loss, as indicated in Figure 1 [17]. The most widely applicable and known coated fertilizers are sulfur-based and mineral-based coated urea. However, due to the high cost, limited resources, and soil ecosystem pollution from both sulfur-based and mineral-based coating materials, the use of organic polymers is preferable due to their abundant quantities, inexpensive cost, and biodegradable, renewable, and environmentally friendly characteristics (Figure 1). According to Fertilizer Europe [18], the European Union regulations set the rules for controlled-release fertilizer such that midway through 2021, the use of microplastics as coating materials for urea was restricted, and from 2026 onwards, only materials that meet the biodegradability requirements will be approved for use as coating materials [18]. The restriction aims to conserve the soil ecosystem and bodies of water from the pollutants associated with non-biodegradable microplastics. 

Starch, a naturally based coating material, is the most-researched natural polymer for coating urea due to its high availability, low cost, and environmentally friendly nature (Figure 1). Rychter et al. [19] prepared a starch-based, controlled-release fertilizer with urea which serves as a plasticizer to reduce moisture content, affecting the mechanical properties and crystallinity of the matrix. The use of starch in urea coating also serves as modifier, binder, or sealant for chitosan, providing the function of reducing the number of pores in the surface of the urea granule [20]. Chitosan–starch coatings increase the water absorption ability [20]. Starch blends with polyurethane and polysulfone have been used with the modification of the ratio of synthetic polymers to natural biopolymers to achieve the desired biodegradability of the coated urea in the soil [21]. In other studies, cassava starch has been proven to be a low-cost coating membrane for improving slow-release fertilizer while having a low negative impact on the environment. However, when corn starch was thermally processed into a coating material using disodium tetraborate and urea and coated over granular urea in a vertical bed coating reactor, the resulting coated urea was reported to have properties such as a uniform, dense, hard, and least-porous surface [22].

Compared with the uncoated urea granules, which released N into water in 6 min under mild shaking, the corn-starch-coated urea took approximately 32 min to completely release N [22]. 

In this review, focus is narrowed to the use of organic materials for urea coating, as summarized in Table 1. The organic materials for urea coating can be used directly, after some modification or mixing with minerals for binding, and for sealant purposes (Table 1). Organic coating materials with a high biodegradability and renewability due to their derivation from natural products encourage researchers and industrialists to search for innovative and sustainable organically based materials that can reduce the cost of coated fertilizer production while providing a high efficiency and superior properties of controlled N release from urea fertilizer. Organic materials such as biochar, resin, starch, and polyphenol (Table 1) are widely used to coat urea granules for the slow release of N, to reduce environmental pollution, and to decrease the harmful effects associated with sulfur-based and mineral-based coatings. Synthetic polymers such as polyethylene, polystyrene, and polyesters are thermoplastic materials used as urea coating materials due to their apparent benefit over natural polymers, such as a set-to-set consistency, foreseeable physico-chemical properties, and tailor-made character (Table 1). Polymer-coated urea (PCU) has great potential for increasing crop production via enhancing N fertilizer use efficiency and benefiting the ecosystem by preventing pollution from the excessive use of urea. However, most PCUs are used only in a limited market due to their high cost compared with conventional N fertilizers. To address the problem of high-cost PCUs, Yang et al. [23] studied a low-cost PCU and a large tablet polymer-coated urea (LTPCU) which were prepared using recycled polystyrene foam and various sealants as coating materials. The recycled polystyrene foam was the ideal coating material for the controlled-release fertilizer. The polyurethane, which was synthesized through the reaction of castor oil and isocyanate, was better than the wax as an additive for delaying the N release rate of the coated urea. The 70–80% less coating material was used for the LTPCU than for commercial PCUs with a similar N release longevity. Moreover, the cost involved in recycling the polystyrene foam to coat one ton of pure N with the LTPCU was approximately one-seventh to one-eighth of the cost of the traditional polymer used for the commercial PCU [23]. For resin-based materials for urea coating, acrylic resin and epoxy resin were used as a protective layer to improve the hydrophobicity of the coated urea for controlled release [24] (Table 1). 

The use of a natural, biodegradable polymer such as starch for urea coating constantly increases because when starch-coated urea is applied to the soil, the degradation begins with the presence of bacteria, fungi, and algae enzymes which catalyze the process of chemical hydrolysis [46]. In addition, starch is naturally synthesized by plants; therefore, starch is completely biodegradable, able to polymerize after some processing, relatively inexpensive, and sustainably available [47,48]. The wide range of starches provides options for starch selection for coating urea granules, depending on the composition of the starch, to obtain the desired properties of the polymeric film. As summarized in Table 1, plasticization is used to obtain a more elastic film, as some studies stated that the urea solubility decreased when it was added into the matrix of a starch-based polymer [49] or when a combined urea coating with starch and polyacrylic acid was used [50].

## 3. Methods of Urea Encapsulation 

Some of the methods adopted for urea coating are pan coater, rotary drum, and fluidized bed coater technology [17]. Among these, most of the studies reported that the fluidized bed technique produced a good coating which improved N release. Fluidized beds, both with and without internals, offer many advantages over conventional granulation and coating equipment such as pans, drums, and mixers. The air or gas required to fluidize the solids, which typically have high rates of heat and mass transfer, lead to a uniform temperature distribution within the bed and relatively short processing times. Moreover, the shearing forces exerted in such beds help to control the formation of agglomerates, and the movement of fluidizing gas, including bubbles, causes solids to circulate within the equipment. This provides a constant flow of bed particles through the spray zone, which is essential for uniform product quality [51]. One of the applications of fluidized bed technology is a fluid bed coater for producing sulfur coated urea. In fluid bed coater, the coating solution can be prepared in several ways, such as by melting and wet and dry coating. The drying process of the fluid bed coater involves exhaust air humidity, and the drying rates are measured by means of a humidity meter. The measurement of humidity is conducted both for inlet air humidity and outlet air humidity. Although most of the studies show fluidized bed coating provides a high yield of coating, the method provides the best results when a correct proportion of the solution is used. Excessive solution causes the granules to be clumped together and leads to agglomeration, which results in the collapse of the bed [17]. 

For the rotary pan operation procedures, urea particles are shaken by the rotation of the pan and sprinkled with solution by the spray nozzles. Throughout the sprinkling operation, high-temperature air is applied over the wet coatings for drying purposes [52]. When the urea granules are exposed to a high air temperature, it might dissolve the urea, thus reducing the quality of the urea [53]. In addition, blowing air is provided to dry the coated urea granules, which come out of the pan due to the rotation. The rotary drum has some drawbacks, such as the rotation increasing the crushing strength which can easily break the granules. Thus, water can easily penetrate and dissolve the urea. To address this problem, polyols could be used to decrease the crushing strength of the coated urea and slow down the release rate. However, polyol is expensive and not affordable for farmers [54]. In addition, these methodologies, which require a special facility, are very expensive and need trained and skillful personnel to operate the machine. The effect of temperature has been known to increase the risk of NH_3_ volatilization from urea [55]. In our work, we used oven drying during urea encapsulation because we chose to use rejected sago starch (RSS), which is produced during the sieving process of sago starch flour. Oven drying is necessary for the dry coating method. In addition, we considered the dry method for encapsulating urea to be convenient for storage and field application. Limited information is available on the use of native starch from sago flour as a coating material for urea encapsulation. The sago industry in Sarawak, Malaysia, is presently dominated by smallholders who account for 92% of the total estimated planted area of 32,329 hectares. There are currently eight sago processing factories in operation, mainly in the Mukah and Pusa (Sarawak) divisions, with an average production of up to 1.5 metric tonnes of sago starch per hour [56]. To complement and reduce the reliance on existing organic materials for urea coating, as summarized in Table 1, a natural polymer such as RSS could be considered an alternative coating material for urea because it is abundantly available, biodegradable, environmentally friendly, hydrophilic, and has a low cost and a weak coating barrier [57]. Apart from these factors, the physico-chemical properties of RSS are the most important.

In our previous work, we coated urea using a self-built mini rotary machine which was assembled from two stainless steel basins with a diameter of 30 cm and a depth of 10 cm that were attached to a driller that rotated the basins. A fan regulator was used to control the speed of the rotary machine. In our study, the method of preparing the coating materials was slightly modified from the method of Himmah et al. [58]. Himmah et al. [58] used a coating solution made from 2% *w*/*v* of starch, carboxymethyl cellulose (CMC), and 1 g of polyethylene glycol (PEG) [58], whereas in our study, the coating solutions were made solely from RSS at different concentrations, using water as binder or sealant to determine the optimized concentration value (s) that could be used for the urea coating material to reduce NH_3_ loss and NO_3_^−^ leaching from the application of urea. Our study confirmed that the lower dissolution rate of coated urea using an RSS concentration of 2% was found to be suitable for encapsulating urea compared with other concentrations of 3%, 4%, 5%, and 6%, respectively. The lower dissolution rate with the lower RSS concentration used for the urea coating material in our study contradicted the findings of Naz and Sulaiman [59]. Increasing the composition and volume of the coating solution increases the N release time. Similar findings on increasing the coating solution composition in line with the release time were also observed in a study of urea coated with a waterborne starch biopolymer [60]. 

## 4. Rejected Sago Starch from Sago Flour Production 

The economic activity of sago starch flour production in Sarawak, a state in East Malaysia, has played an essential role in sustaining rural livelihood, either for subsistence, trading at the local and domestic market, or for export to Taiwan, Japan, China, and Korea. As shown in Figure 2, Sarawak is the leading exporter of sago starch flour, exporting about 25,000–40,000 tons of sago products annually to Peninsular Malaysia, Japan, Taiwan, and Singapore [61].

The sago segments are commonly immediately transported to the mill for sago flour processing to ensure the premium quality of the sago starch and to avoid deterioration of the starch in the trunk. Sago palms are harvested for starch at the age of 7–15 years, when the starch fully developed in the sago stem and before flowering [63]. The stem of one well-matured sago palm can be segmented into 10–12 logs, which produces 150–400 kg of starch [64]. Starch from a sago trunk is extracted by splitting the stem lengthwise and removing the pith, after which it is crushed and kneaded to obtain the starch before being washed and strained to extract the starch from the fibrous residue. The suspension of raw starch in water is then collected in a settling container. Sago starch contains 27% amylose (the linear polymer) and 73% amylopectin (the branched polymer) [65,66,67]. Flach [68] estimated that the difference in amylose content might occur according to the age, variety, or growing conditions of the sago palm. During starch extraction and after pith pulverizing, the high-quality starch is separated from the rejected starch by sedimentation, which involves rotary sieves, extractors, and sieve bends. Low-quality sago starch is considered rejected starch, which is expelled from the rotary sieves, extractors, and sieve bends. The nature of RSS is almost similar to that of high-quality starch flour except for the texture of RSS, which is slightly coarse (it does not meet the standard sieve for premium quality) (Figure 3). The sedimentation of refused starch during flour extraction is to ensure the high quality of the sago starch for export purposes and domestic consumption. 

### Physico-Chemical Properties of Rejected Sago Starch

In our previous study on coating urea granules with RSS, we analyzed the selected physico-chemical properties of RSS, as summarized in Table 2. The pH of RSS in distilled water was 3.43, which is acidic (Table 2). The organic matter and organic carbon of RSS were high due to the presence of six carbon atoms. When sago starch decomposes, it increases the soil total organic matter and organic carbon, which also increases the soil cation exchange capacity (CEC). As the soil CEC increases, it can retain more positively charged ions, particularly NH_4_^+^. Additionally, RSS has a high number of basic cations, especially K^+^ (170.27 cmol_(+)_ kg^−1^) and Mg^2+^ (20.3 cmol_(+)_ kg^−1^), and its inherent basic cation content helps to enhance its CEC, suggesting an improvement in soil fertility and nutrient retention [69]. Both the swelling and water solubility properties of sago starch suggest an interaction between the amorphous and crystalline areas [70]. In addition, it is influenced by the characteristics of amylose and amylopectin [71]. Sago starch has a high swelling power and a low solubility because of the lower amylose content (Table 2).

Moreover, the unique properties of sago starch, such as hydrophilic nature, enable it to dissolve in water. Although sago starch is hydrophilic, it is highly water sensitive, which can prevent water from entering into the starch film, thus slowing down the hydrolysis of urea by preventing water from reacting with the urea granule. Hydrophilic polymers are characterized by the presence of a main polymer “backbone” with attached polar groups, and they contain micropores. The presence of abundant hydrophilic groups (sago starch contains a hydroxyl group) and micropores in the starch results in the protective layer being less resistant to water and easier to crack, enabling the dissolution of the nutrients. The presence of functional groups can help to retain a positively charged ion (NH_4_^+^), which is then taken up by the plants. In addition, micropores allow small molecules such as NH_4_^+^ and NO_3_^−^ to diffuse through the hydrogel [72]. Other than that, RSS is suitable for coating urea because of its stickiness, which is characterized by the amylose and amylopectin in the starch. Amylopectin is an extensively branched structure and is composed of short chains of α-(1,4)-linked D-glucosyl units that are interconnected through α-(1,6)-linkages [73]. The high content of amylopectin in RSS is responsible for its stickiness. When sago starch is sticky, it can serve as a binder; thus, when starch is used as a coating material, it does not require the addition of a plasticizer/ binder to improve the mechanical strength of the coating. Coated urea is considered a controlled-release fertilizer that releases available forms of N (NH_4_^+^ and NO_3_^−^) in a delayed manner that is in synchrony with the needs of crops for NH_4_^+^ and NO_3_^−^, thus improving the N use efficiency with an enhanced yield. As sago starch is a natural, hydrophilic polymer, the movement of N from the polymer into the surrounding soil is driven by a mass transfer resulting from a concentration gradient and by Brownian molecular motion (self-diffusion). In this way, the diffusion of nutrients from the polymer is similar to that of the diffusion from other inorganic sorbents used as release-controlling devices. Thus, the result of the restricted diffusion of nutrients from within the polymer can delay the release of nutrients into the surrounding soil. 

Rejected sago starch has a moisture-retention property which facilitates absorption, stores a large quantity of water, and enables the water absorbed to be released slowly when the soil dries. The swollen coating that remains in the soil for some duration will serve as an additional reservoir for the plant–soil system, which can prolong the irrigation cycle and also strengthen the ability of plants to tolerate drought [74]. Using RSS as coating material for urea is an ideal controlled release because it is a natural and environmentally friendly macromolecule that retards N release to such a slow pace that application can be optimized, which results in improvements in crop growth, development, and yield [12]. 

The morphology of uncoated urea and RSS-coated urea were analyzed under scanning electron microscope (SEM), which showed that the uncoated urea granule surface was rough, with fine openings and a porous structure through which water can penetrate to dissolve the urea granule. On the other hand, a decrease in membrane porosity and uniform layer were observed in surface morphology of the RSS-coated urea (Figure 4). The RSS encapsulation of the urea granule slowed the penetration time; thus, the RSS-coated urea will take a longer time to dissolve and gradually desorb N to the outer surface of the urea. This characteristic is important for increasing the encapsulation efficacy of the released N. In addition, the acidic or alkaline nature of the release medium has a significant effect on the interactions of chemical species in the urea granule and the diffusion coefficient of the ions [75]. As summarized in Table 2, the pH of RSS is acidic. Under an acidic condition, which is at a pH 2–5, there is high concentration of H^+^ ions, leading to the protonation of the carboxylate anions (COO^−^) and preventing anion–anion electrostatic repulsion in the network, decreasing the swelling capacity. In contrast, the presence of Na^+^ ions in the solution shields the COO^−^ anion and prevents anion–anion electrostatic repulsion in an alkaline environment (>pH 9). Thus, the pH of the coating material is important, as it can acidify the soil microsite and reduce the swelling capacity. In our study, coating urea with RSS minimized the emission of NH_3_ because the acidity of RSS, which is 3.43, enables the urea granule surface to be temporarily acidified. In addition, when the pH is less than 5.5, urea hydrolyzes slowly [76,77]. Consequently, the lower pH effectively increased the time required for complete urea hydrolysis.

## 5. Mechanism of Nitrogen Release from Coated Urea

The mechanism of the slow release of N from coated urea reflects the effectiveness of a slow-release fertilizer and the fertilizer’s efficiency. However, the slow release of N from coated urea is complicated to visualize because the release mechanism of N from coated urea depends on several factors, such as the nature of the coating material, atmospheric temperature, soil moisture, agricultural environment, farming methods, and many other variables. As summarized in Figure 5, after applying the coated urea to the soil, water penetrates the coating to condense on the solid fertilizer core, followed by partial N dissolution. 

After the release of NH_4_^+^ in the soil, Figure 6 depicts the availability of NH_4_^+^ for plant uptake, leading to increase in the NUE and minimizing the frequency of N fertilizer application. In addition, the coated urea mitigates N loss through the volatilization of NH_3_ and the emission of nitrous oxide (N_2_O) in such a way that the coating serves as a nitrification inhibitor, which reduces the rate of nitrification of NH_4_^+^ to NO_3_^−^. Thus, N_2_O and nitric oxide (NO) emissions, NO_3_^−^ leaching and/ or run-off, and the denitrification of NO_3_^−^ are reduced during this process. The mechanisms of N release from the coated urea take place in three different stages, such as the lag, linear, and decay regions, while the uncoated urea follows a linear trend (Figure 6). In the lag region, water in the soil penetrates the coating and a small fraction of urea fertilizer is dissolved. The vapor pressure gradient is the driving force; thus, urea hydrolysis is delayed at the lag stage (Figure 6) [78]. The thickness of the coating also affects the rate of N release from coated urea, as Shaviv et al. [79] reported that the coating thickness is proportionate to lag period and is inversely proportional to the release rate in the linear and decay phase. The study suggested that by increasing the coating thickness, the lag period can be prolonged, and the release rate can be slowed down in both the linear and decay phases. Increasing the coating thickness is generally preferred in the interest of economic feasibility but in some cases, it depends on the properties of the coating material.

In the linear region, soil water keeps penetrating to react with the urea; thus, the osmotic pressure in the core builds up, and the critical water volume of the saturated solution accumulates, which enables the N from the urea to be slowly released through the cracks in the coating. As the concentration of the solution inside the granule remains saturated, the diffusion to the soil is constant [80]. If the pressure exceeds a prescribed threshold value, the pressure will facilitate the rupture of the coating material and the immediate burst release of the fertilizer content. In the decay stage, most of the urea has been dissolved and released, which reduces the concentration gradient and the driving force and thus the release rate. The pattern reflects that the coated urea samples release slowly over time at the start and later times and more quickly in the middle periods [78] (Figure 6). 

Figure 7 shows the N retention in soil, whereby N retention can be defined as the soil’s capacity to retain inorganic N (NH_4_^+^ and NO_3_^−^) against losses through mineralization. In the early stage of incubation, uncoated urea dissolved rapidly due to its high solubility and the rapid urease-catalyzed hydrolysis of urea, whereas RSS-coated urea demonstrates a delayed solubility due to the presence of the physical barrier of the coating layer; thus, N loss is lower than for the uncoated urea (Figure 7). Most of the N from the uncoated urea is consistently released, whereas for the RSS-coated urea, the mechanisms of N release occur when the soil water reacts with the coating layer, followed by the swelling of the coating layer and the gradual solubilization of the coating material (RSS). Thus, N is eventually released within the incubation period, which is from day 40 to day 120 (Figure 7). At 80 days of incubation, all the N from the uncoated urea was completely dissolved, and there was a lower soil total N concentration, whereas the RSS-coated urea released a higher amount of N into the soil (Figure 7). Compared with the uncoated urea, N is retained in soil with the RSS-coated urea due to the preservation of N during the early phase of incubation, which is at 40 days of incubation. In addition, the use of RSS, which has an acidic nature, leads to a stronger ability to adsorb N in the form of NH_4_^+^. As a result, the RSS-coated urea retained N in the soil when compared with the uncoated urea (Figure 7). The slow N release pattern of the RSS-coated urea, as indicated in Figure 7, suggests the retention of N in the soil, ascertaining the availability of N for crop uptake.

## 6. Drawbacks of Natural Coating Materials

Although natural materials are renewable, biodegradable, and environmentally friendly, some organic coating materials have disadvantages, as summarized in Table 3. The main drawback of using some of the natural coating materials listed in Table 3 are that they are costly, exhibit poor biodegradability, and have weak thermophysical and mechanical strength. Thus, those materials remain in the soil and disturb the soil ecosystem.

The production of slow-release fertilizers with biodegradable coatings, such as the use of RSS, is a huge contribution to soil ecosystem and organic farming. To achieve fully organic farming, other native starches, such as potato starch, were used as a urea coating. A single layer of native potato starch substance for urea coating increased the static strength of the urea granules by up to 22.7 N/granule, while a two-layered coating reduced the granules’ strength to 19.3 N/granule but was still higher than the minimum requirement provided by fertilizer producers, which is 7.0 N/granule [86]. The mechanical properties of the double coating of native potato starch are insignificantly different to the single coating, as both coating layers reduce the solubility of the urea by more than 20% [86]. Azeem et al. [87] stated that due to their poor mechanical and thermophysical properties, natural polymers such as starch are not suitable for urea coating unless they are modified with some appropriate crosslinker, plasticizer, or stabilizer. Despite the chemical or physical modification of starch for urea coating, the starch could not yield promising results in terms of effective controlled release characteristics, cost of production, viability for scaled-up production, and ease of application of the native-starch-coated urea. Different starches were modified and mixed with borax, polysulfone, and polyvinyl alcohol/boric acid and used as coating materials for urea encapsulation. However, the N release time from starches mixed with borax, polysulfone, and polyvinyl alcohol/boric acid was as low as 0.05, 5.0, and 8.0 h, respectively. The studies on using native starch are still ongoing and require extensive improvement in their coatings’ thermophysical and mechanical strength. 

To that end, this paper reviewed the potential of RSS, the solid waste from sago flour processing, as a coating material for urea encapsulation. The proper application and better understanding of the function of RSS as coating material could improve the retention of soil-available inorganic N by controlling the rapid release of N during urea hydrolysis due to the specific physicochemical properties of the RSS. Inorganic, organic, and natural polymers have been widely explored for the formulation of urea encapsulation with the properties of slow-release fertilizers or controlled-release fertilizers. In addition to its function as a natural polymer for urea encapsulation, starch is also used as a binder and filler. Starches that have been modified for their thermophysical and mechanical strength are very important in the release pattern because each starch has specific properties that can facilitate the desorption of N from the coated urea. The existing materials for urea coating are not necessarily the best options because they have some drawbacks, such as being expensive and non-biodegradable. This review revealed the potential of RSS as a urea coating material for agricultural means and highlighted the benefits of RSS, which is abundantly available, low-cost, and considered waste in sago starch flour processing. In terms of agronomic efficiency, the coating method is easy and does not require any expensive equipment. Thus, the adoption of the RSS-coated urea by farmers could be affordable. From economic and environmental efficiency perspectives, the use of RSS is considered low-cost because RSS is waste generated from sago starch processing that has the potential to pollute the environment.

## Figures and Tables

**Figure 1 polymers-15-01863-f001:**
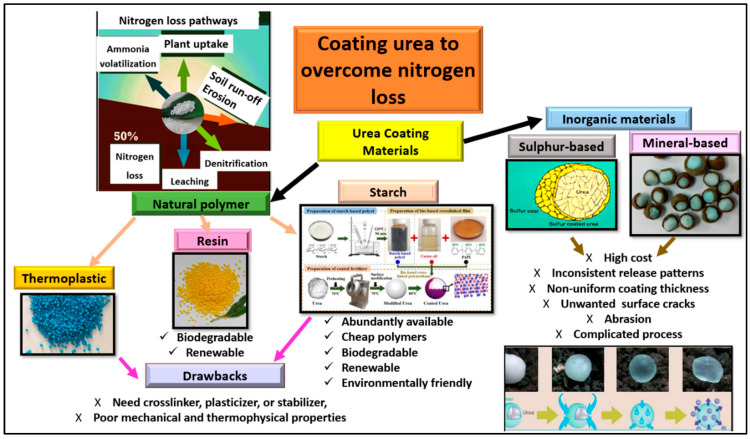
Common materials used for coating urea granules to prevent rapid nitrogen loss from urea.

**Figure 2 polymers-15-01863-f002:**
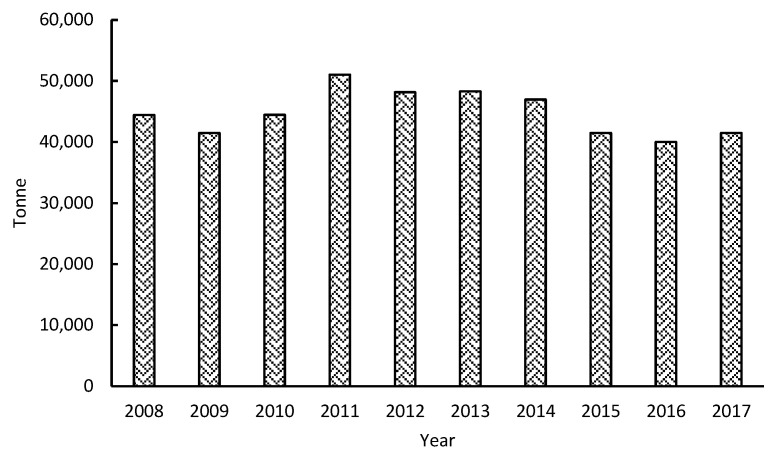
Ten years of sago starch export statistics for Sarawak, Malaysia, 2008–2017 [62].

**Figure 3 polymers-15-01863-f003:**
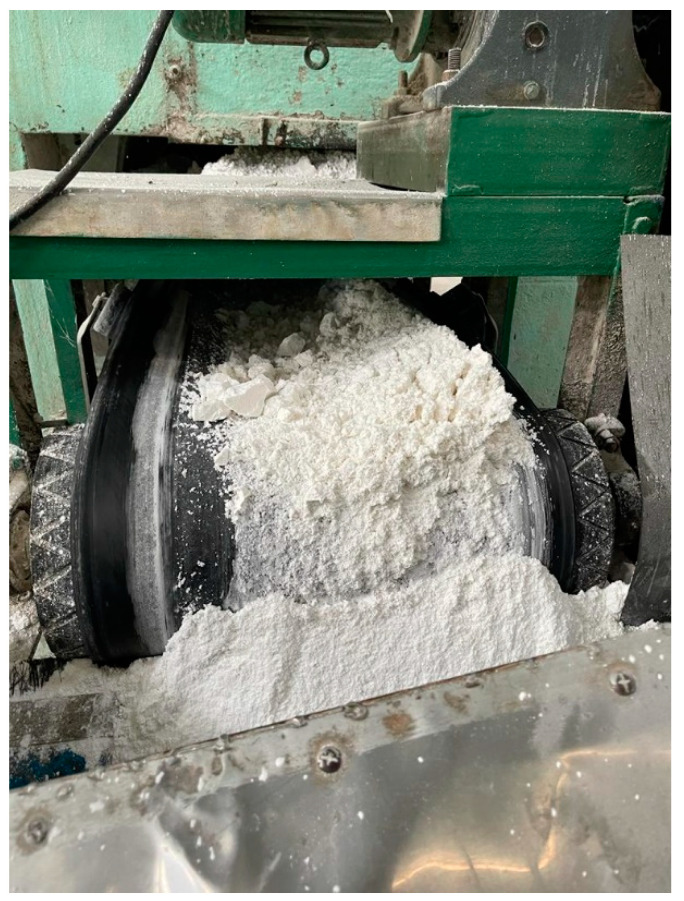
The rejected sago starch, which did not meet the requirement of premium quality for domestic and export purposes.

**Figure 4 polymers-15-01863-f004:**
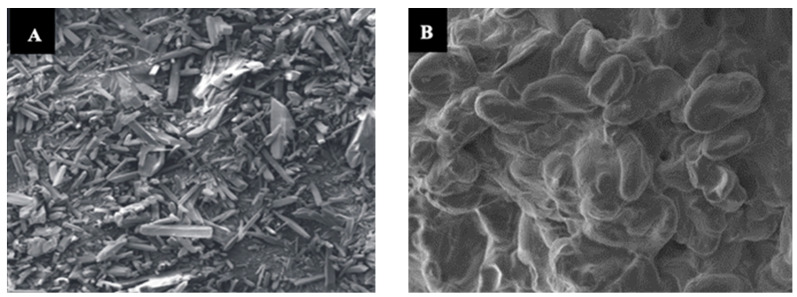
Surface morphology of uncoated urea (**A**) and rejected-sago-starch-coated urea (**B**).

**Figure 5 polymers-15-01863-f005:**
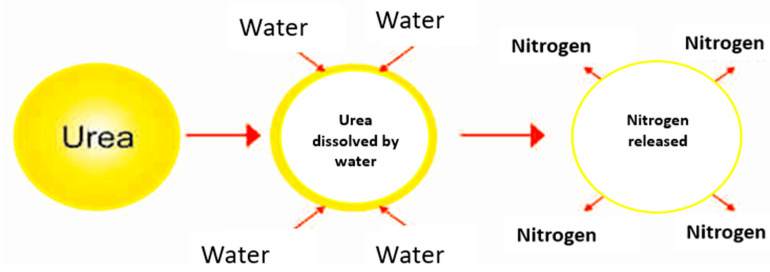
Diffusion mechanism of nitrogen released from coated urea.

**Figure 6 polymers-15-01863-f006:**
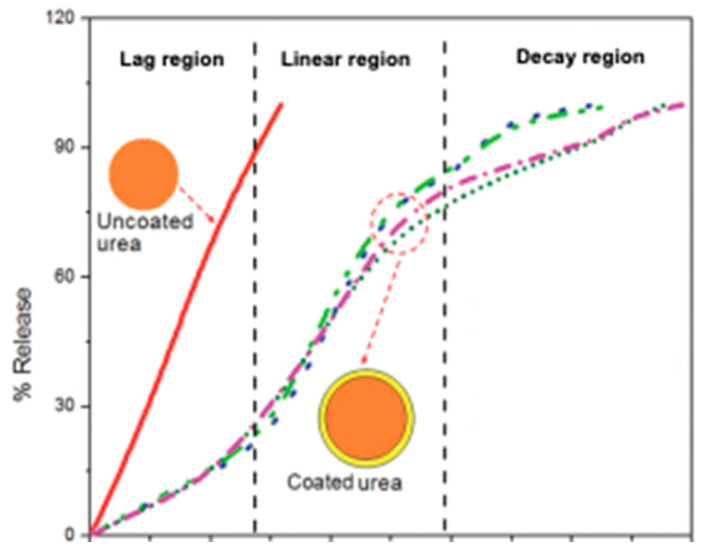
Nitrogen release pattern of uncoated urea and coated urea [78].

**Figure 7 polymers-15-01863-f007:**
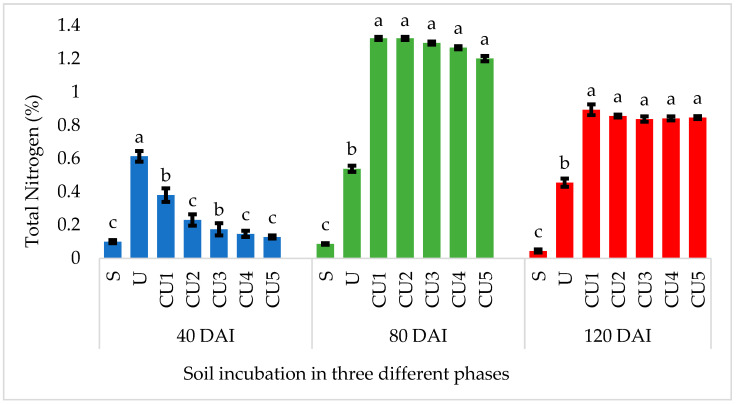
Nitrogen released from uncoated and RSS-coated urea in soil incubated for 40, 80, and 120 days. Note: S—soil only; U—uncoated urea; CU1, CU2, CU3, CU4, and CU5—urea coated with different concentrations of rejected sago starch [57]. Means with different letter(s) within the same incubation period indicate a significant difference between treatments, determined by Duncan’s new multiple range test (DNMRT) at *p* ≤ 0.05. Bars represent the mean values ± SE.

**Table 1 polymers-15-01863-t001:** Organic materials commonly used to encapsulate urea granules.

1. Thermoplastic-Based			
Material	Modifier/Binder/ Sealant	Research Findings	References
Poly (butylene succinate) (PBS)	Empty fruit bunch	Irradiation of PBS prior to mixing improved the adhesion and increased the biodegradation rate compared with the non-irradiated SRF composites. This is because EFB fibres are hydrophilic in nature, while the matrix polymer is hydrophobic. The mixture of these two materials causes weak adhesion and poor fibres.	[25]
Poly (lactic) acid	-	Utilizing the higher molecular weight poly (lactic acid) resulted in a slower urea release due to the decreasing permeability of the PLA. This slow-release fertilizer is promising because it does not leave residues that damage the soil structure or the nutrient balance in the soil.	[26]
Polyester	-	The increasing size of the controlled-release fertilizer while using smaller urea crystals slows down the degradability and release rate.	[27]
Polyurethane	-	The application of at least 50% total N as coated urea strongly reduced N leaching and improved N agronomic efficiency in comparison with traditional fertilizers, ensuring a similar fruit production in the same time.	[28]
	Mesoporous silica	Filler morphology affects the release rate.	[29]
	Hydroxypropyl-terminated polydimethylsiloxane (HP-PDMS)	Implementation of hydrophobic gradient layer increases urea diffusion resistance.	[30]
Polystyrene	Wax, polyurethane	Wax is brittle and cannot prevent water penetrating the coating. Increasing the size slows down the release and reduces the amount of coating material required.	[23]
Polyether sulfone	Fe_2_O_3_ nanoparticles (NPs)	A new class of controlled-release fertilizer. Fe_2_O_3_ NPs increase the coating thickness and reduce the release rate. They also allow the carrier to be recovered and recycled.	[31]
**2. Resin-Based**			
Acrylic resin	N-(n-butyl) thiophosphric triamide(NBPT) and dicyandiamde(DCD)	The film is integrated on the urea core surface, the coated material is uniformly distributed with the coating and closely combined with the urea core, and the surface is smooth and able to control the urea release.	[32]
Bio-based epoxy	-	Urea coating with epoxy is environmentally friendly and has a superior controlled release property.	[33]
**3. Natural-Based**			
Cellulose	Silica NP, bentonite, montmorillonite (MMT)	Incorporation of filler into cellulose-based coating material promotes tortuous path and compactness which slows down diffusion.	[34]
	Ethyl cellulose (EC) as inner coating and cellulose-based superabsorbent polymer (cellulose-SAP) adsorbing biochemical inhibitors dicyandiamide (DCD) and thiourea as outer coating.	It has the functions of water retention and slow release, but it also inhibits the conversion of NH_4_^+^-N to NO_3_^−^N.	[35]
Lignin	Alkenyl succinic anhydride	Film-forming properties show great potential to retard nutrient release.	[36]
	-	Lignin indicated as an eco-friendly material with good controlled-release capacities that potentially could be applied in agriculture and horticulture.	[37]
	Phenol-formaldehyde resin	Phosphorus is partly retained inside the slightly soluble calcium sulphate matrix.	[38]
	Acetic acid/sodium metabisulfite	Lignin increases the efficiency of the delivery of the bioactive material for a longer period and prevents the pollution of surface and underground water with NH_4_^+^.	[39]
Chitosan	Magnesium, rice-husk-ash, liquid natural rubber, Epsom salt	Increases the yield of rice.	[40]
	-	The chitosan’s resistance slows down the release rate of N, P, and K.	[41]
	Poly (acrylic acid-co-acrylamide) (P(AA-co-AM)	Properties of chitosan-coated NPK compound fertilizer with controlled-release and water-retention features releases the nutrient slowly.	[42]
	Starch	The presence of chitosan on the coating system is responsible for producing a porous matrix surface, while the availability of starch tends to reduce the number of pores in the surface. The swelling analysis shows that the presence of chitosan–starch coatings increases the water absorption ability.	[20]
Biochar	Bentonite	Water retention and controlled-release properties improve the N utilisation efficiency and reduce the environmental impact.	[43]
	-	Biochar with a high specific surface area, hydrophilic oxygen-containing functional groups, and a low pH slow down the release.	[44]
	-	Porous nature and surface functional groups of biochar minimize NO_3_^−^ leaching and improve the NUE.	[45]

**Table 2 polymers-15-01863-t002:** Selected physical and chemical properties of rejected sago starch.

Property	Rejected Sago Starch
pH_water_	3.43 ± 0.050
EC (µS cm^−1^)	1459.00 ± 58.000
Total organic carbon (%)	43.05 ± 2.666
Organic matter (%)	74.23 ± 4.596
Exchangeable NH_4_^+^ (ppm)	3.74 ± 0.234
Available NO_3_^−^ (ppm)	2.34 ± 0.467
Available P (mg kg^−1^)	0.87 ± 0.065
Exchangeable K^+^ (cmol (+) kg^−1^)	170.27 ± 68.311
Exchangeable Ca^2+^ (cmol (+) kg^−1^)	0.78 ± 0.020
Exchangeable Mg^2+^ (cmol (+) kg^−1^)	20.13 ± 0.811
Exchangeable Fe^2+^ (cmol (+) kg^−1^)	0.41 ± 0.064
Exchangeable Na^+^ (cmol (+) kg^−1^)	0.12 ± 0.062
Swelling power (g/g)	10.0 ± 0.153
Solubility (%)	0.56 ± 0.006

Note: Data extracted from Kavitha et al. [57].

**Table 3 polymers-15-01863-t003:** Drawbacks of natural existing materials.

Natural Existing Materials	Drawbacks	References
Biochar	Expensive due to the high cost of collecting and obtaining the feedstock, as well as the high capital, operating, and maintenance costs of the pyrolysis plant.	[81]
Cellulose	Raw cellulose materials are poor in mechanical strength and have fast release rate; thus, chemical modifications are commonly performed to improve their properties.	[82]
Lignin	Due to the different sources of lignin and the different extraction methods, the structure of technical lignin varies greatly. Therefore, when it is used as a carrier for slow-release fertilizer, the different physical and chemical properties of lignin with different structures need to be considered.	[83]
Chitosan	Must be combined with various polyanions from natural polymer to improve the physical and chemical properties. Is usually soluble in acid. For chitosan to be soluble in water, it requires expensive preparation, which would result in an expensive fertilizer and therefore raise the chances of market refusal.	[84,85]

## Data Availability

The data are available in a publicly accessible repository.

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
