# Peer review of "Potential of Rejected Sago Starch as a Coating Material for Urea Encapsulation"

_polymers, 2023, doi:10.3390/polym15081863_

Round 1

Reviewer 1 Report (Previous Reviewer 3)

Dear authors,

your manuscript has been improved again, but it is not well organized. I suggest to compile all information on sago starch and its potency to be used as a coating material is presented in a last chapter, while the general information needed to understand this potency should be presented before. There is no need for conclusions, as there are no data to derive conclusions from.

Use of English language (in particular grammar) has to be checked throughout the document

L 23 application of NBPT should not be classified as coating

L 59 losses of N can occur under any favourable conditions for these processes and are not necessarily an indicator of excess fertilization. Losses can cause N limitations.

L 66 not necessarily, when no N loss occurs, applied N is readily available for the crops. That’s why straight n fertilization works in most cases.

L 75 which can result in (see points above)

L 82ff only if the coating works effectively in the desired way. This statement is not correct for all kind of coatings as such.

L 124 f do not agree, excessive fertilization can also be done with encapsulated fertilizers, causing similar environmentral problems. Fertilizer form is not a solution per se but depends also on proper application.

L 145 ff good point, it could be mentioned that there exists a regulation om micro plastics in the EU which does not allow the application of non-degradable micro-plastics with fertilizers.

L 154 costs need to be generally lower for renewable, biodegradable coatings. Please show also shortcomings of existing natural based coatings (i.e. robustness of coating effect etc.) and why there is a need for another more effective natural based coating material.

Chapter 3: available product potential could be also be included in the chapter and chapter heading, nevertheless this section is still very long and could be shortened.

L 282 what is the potential amount available?

Chapter 3.2. What about the robustness of the slow release characteristics, can sago starch be used to produce a uniform product with simiar release behaviour across soils and environmental conditions? All the writing is very hypothetic. Which additional research and information on sago starch coating behaiour is needed to create a coating product for the fertilizer industry?

Parts of Chapter 4 eventually better before chapter 3, and connect sago starch related coating technology to information on sago starch

L 408 ff. why, give data.

Chapters 4.2, 4.3 and 5: not sago starch specific should be moved up, same for chapter 4.3 and 5. General information before information on sago starch.

L 496 not minimized, but reduced

Fig 8 probably better to show just the principle without distinct indication of time in the x-axis, because the dynamic behaviour depends on coating characteristics, material, environmental conditions etc.

L 546 ff this should be part of the sago chapter.

Fig 9, much too simple, vertical dashed line needs explanation, indicate measure of variation (error bars), explain the term N retention in soil. The explanation of the graph is very hard to understand.

Conclusions can be merged with the sago chapter, as there is nothing to conclude from the data. Actually only very limited evidence of the efficacy of sago starch coating are presented. Only the potency of this material is supported which calls for much more experimentation and evidence.

Author Response

Reviewer 2 Report (Previous Reviewer 2)

Reviewer comments-R4

Polymers- MSPI  

Manuscript ID: polymers-2292723

Manuscript Title: Overview on the potency of rejected sago starch as coating material for urea encapsulation"

Dear Respected Editor-in-Chief,

After reviewing this review article for the fourth time, I find that it has improved and became publishable, as previously mentioned in a previous review, but after taking the following comments into consideration:

1- Modified the title to Overview of the potency of rejected sago starch as a coating material for urea encapsulation"

2- Convert Figure (1) to a graphical abstract after making the appropriate modification to it so that it becomes a clear graphical abstract that expresses all the content of this review article

3-     Figures (2) and (4) have been completely omitted from that review article, as it is not needed.

4- Review all of the abbreviations throughout that review article.

5- References must be checked, all scientific names must be written in italics, and the first l (azadirachta indica, Metroxylon sagu Rottb.  etc., references must be updated, and the first letter of the gender name should be capitalized and pls focus on the last five years, especially 2021-2023, please refer to the journal guidelines.

6- Writing in the references was done in different fonts. The type and size of the font must be unified according to the instructions of the journal.

7- Referance No. 13 “N2O” replace by N2O.

8-Thoroughly check the typos, syntax errors, and appropriate uses of full forms and their abbreviation.

Thanks again and I am ready for any future cooperation with your wonderful Journal.

Regards,

Esmat Farouk Ali Ahmed

Round 2

Reviewer 1 Report (Previous Reviewer 3)

The authors have done a lot of changes following my suggestions. The manuscript can now be accepted for publication.

This manuscript is a resubmission of an earlier submission. The following is a list of the peer review reports and author responses from that submission.

Round 1

Reviewer 1 Report

Based on the author's revision, this article can be accepted for publication.

Reviewer 2 Report

Reviewer comments

Polymers- MSPI  

Manuscript ID: polymers-1748913

Manuscript Title: Loss pathways and remediation of nitrogen through encapsulating urea with rejected sago starch"

The new title: Overview on the potency of rejected sago starch as coating ma- 2 terial for urea encapsulation

After revising the attached file containing the response to my previous comments on the this review article, I found that the authors had improved the manuscript and that they responded not completely to most of the comments and questions to be answered. With the exception of some previous comments and a careful linguistic review of the manuscript, I think that it can be accepted for publication.

The reviewer comments:

Abstract:

·        It is preferable if you can add graphical abstract so that it is easier to understand and clearer.

·        Please write a sentence explaining the final recommendation of this review article at the end of the abstract section.

                     Introduction:

·        Pls delete Table 1. There is no need for the presence of this table. The information in Table 1 is known. 

·        We care about this information in the industrial part only, but here it is assumed that the urea is available and available and you are studying its effects after encapsulation

·        The usage of abbreviation should be used after the full term. Please be consistent with the usage of all abbreviations. Pls revise the abbreviations in the whole part of this article.

·        The introduction part should be written, including separate parts, on first the urea as a source of N, second: documented scientific information about the importance of encapsulation used, because what is available from them is very simple information and does not meet the requirements. Third: The purpose of this review article is in clear sentences.

·        Please add relevant ref. in the entire review article especially regarding the scientific information.

·        Figure 1. Type of materials use for coating urea, pls put this figure in the right position

·        Figure 3; Not found in the manuscript

·        Figure 4. Pls add the source.

·        Figure 5. Pls add the source.

·        There are much recent scientific information (2020-2022) that should have been written to support this review article and increase the quality

·        Please add more discussion for more explanation especially in the important section (encapsulation) as much as possible, taking into account the lack of violation of the content and the non-repetition.

Conclusions In conclusion, you should write a summary of the current work in clear and short sentences so that I, as a reader of the review article, can understand what the article ended up being.

·        There is another important part that must be added at the end of this review article, which is the current situation of using this plant and its production in safe form and adding the vision or future expectations of using this plant or its extracts in order to increase the productivity as well as from an economic point of view and in terms of its impact on protecting the environment, ...etc.

                      References:

·        Pls, revise the Ref. carefully.

Reviewer 3 Report

Dear authors,

the topic oft the paper is still highly interesting and relevant. However, the given data and evidence for the use of rejected sago starch is very limited and in some respect highly speculative. There is a strong imbalance between presented scientific evidence and the length of the paper. I suggest to write a paper which suggests to the science community to explore the application of rejected sago starch in more detail for fertilizer coating based on the presented physico-chemical characteristics and compliance with restrictions for coating materials in the agriculture space. The paper by far not acceptable in its current form. I suggest to shorten it to at maximum the half of its length and publish it as a short communication or opinion paper. The amount of data collected on rejected sago starch is so limited that it cannot be considered a review paper.

I abstain from detailed comments unless the paper has been well organized.

Thorough English language revision for the whole document is required.

Section 3 by far too long: use of non-rejected sago starch is not essential, physico-chemical behaviour of rejected starch is of interest and why those make sago starch a relevant biopolymer for coating urea granules. Also harvest etc. not essential here. Eventual negative side effects of sago starch use are relevant.

L 262 – 292 very limited scientific literature or own research giving evidence for the statements, this is not acceptable. The passage highlighted in blue should be shortened by about 50%.

L 327-345 there is no need for this passage in this elaboration on the usefulness of sago starch as coating material and its application.

L 373 ff in this passage, it is not clear to which statement the authors drive at with the given information.

L 386 ff first the authors write about swelling than pH effect: again the statement is very opaque and equivocal.

Passage 4.2. can be omitted as not being relevant for the topic of the paper.

L 408 ff so what does this mean for the topic of the paper, please be more concise?

Fig. 8 Please correct in graph ‘linear region’

Fig. 9:This is just one curve, what about more experimentation under other soil, water and temperature conditions? Such additional testing is essential to understand the coating behaviour. In addition: what kind of coating behaviour is envisaged: short term for ammonia emission suppression or long term for slow release urea. What is the expected length active time of the coating?

Round 2

Reviewer 2 Report

Dear Respected Editor-in-Chief,

After a comprehensive review of the entire review article and tracking researchers' responses to my comments, I want to tell you that this review article in its current form is suitable for publication in Polymers

Thanks again and I am ready for any future cooperation with your wonderful Journal.

Regards,

Esmat Farouk Ali Ahmed

Reviewer 3 Report

Dear authors,

I am in strong disagreement to the other reviewers and I see no improvement of the manuscript in the course of the review process. I therefore do not want to repeat my points of concern and suggest rejection of the paper.